# Eigenvalue-based entropy in directed complex networks

**Yan Sun**[1,2,3], **Haixing Zhao**[1,3]*, **Jing Liang**[1,3], **Xiujuan Ma**[1,3]

**1** School of Computer, Qinghai Normal University, Xining, China, **2** School of Computer, Qinghai Nationality University, Xining, China, **3** The State Key Laboratory of Tibetan Intelligent Information Processing and Application, Xining, Qinghai, China

☯ These authors contributed equally to this work.
* h.x.zhao@163.com

**Data Availability Statement:** All relevant data are within the paper and its Supporting information files.

**Funding:** Project supported by the National Natural Science Foundation of China (Grant Nos.11661069,61663041), the Science and

## Abstract

Entropy is an important index for describing the structure, function, and evolution of network. The existing research on entropy is primarily applied to undirected networks. Compared with an undirected network, a directed network involves a special asymmetric transfer. The research on the entropy of directed networks is very significant to effectively quantify the structural information of the whole network. Typical complex network models include nearest-neighbour coupling network, small-world network, scale-free network, and random network. These network models are abstracted as undirected graphs without considering the direction of node connection. For complex networks, modeling through the direction of network nodes is extremely challenging. In this paper, based on these typical models of complex network, a directed network model considering node connection in-direction is proposed, and the eigenvalue entropies of three matrices in the directed network is defined and studied, where the three matrices are adjacency matrix, in-degree Laplacian matrix and in-degree signless Laplacian matrix. The eigenvalue-based entropies of three matrices are calculated in directed nearest-neighbor coupling, directed small world, directed scale-free and directed random networks. Through the simulation experiment on the real directed network, the result shows that the eigenvalue entropy of the real directed network is between the eigenvalue entropy of directed scale-free network and directed small-world network.

## Introduction

In recent years, research pertaining to complex network topologies has garnered significant attention. The understanding of network topology knowledge is related to the study of network composition, function, and evolution. Many indicators reflect the features of a complex network topology, such as network node degree, clustering coefficient, density, and network diameter, etc [1–3]. Although these indicators can partly reflect the features of the network topology, they cannot fully describe the whole and dynamic characteristics of the network. Insufficient information for mapping the overall network topology is a concern to scholars. In this regard, network entropy [4] is a crucial method. It was proposed by Shannon [5] and

Technology Plan of Qinghai Province, China(Grant No.2019-ZJ-7012).

**Competing interests:** The authors have declared that no competing interests exist.

derived from the information content. Entropy is essential for applications in information science, computer science, statistics, chemistry, astronomy, and other fields [6–8]. The definition of entropy differs in these fields. Among them, graph entropy, also known as network entropy, can describe the node relationship structure. Graph entropy has been widely investigated over the years.

## Problem model

Graph entropy [9] is a theoretical method used to quantify the complex performance of graphs. In 1955, Rashevsky [10] first proposed the concept of graph entropy, which is based on the vertices symmetric structure of chemical molecules. In 1956, Trucco [11] published an article based on the entropy of molecular edges symmetry. In the literature [9], the entropy $I_g(X)$ of an undirected graph $X$ is also given from the perspective of group theory.

Let $A_i$ be the orbits of group $G(X)$, and $p_i = \frac{|A_i|}{n}$, $1 \leq i \leq h$. Then the structural information content is

$$I_g(X) = -\sum_{i=1}^{h} \frac{|A_i|}{n} log \frac{|A_i|}{n} = -\sum_{i=1}^{h} p_i log p_i.$$

The definition of entropy proposed above comes from different fields. Graph entropy based on the in-direction of node connections is challenging for directed complex networks. Compared with an undirected network, a directed network come down to a special asymmetric transfer. It is more difficult to research graph entropy in directed networks than in undirected networks.

## Related work

Graph entropy can be classified based on the invariance of the graph as follows:

1. Degree-based vertex entropy [4], which evaluates the robustness of the network and measures the importance of the vertex. In the protein interaction network, it is used as an indicator to determine the protein contribution.

2. Distance-based entropy, Bonchev and Trinajstic [12] proved that it is more sensitive than other classical topological indicators in mathematical chemistry.

3. Subgraph-based structure entropy, Konstantinova and Paleev [13] described the information metric of a subgraph, it is useful for investigating the overall properties of the graph.

4. Eigenvalue-based entropy [14] derived from entropy defined by Renyi [15].

Eigenvalue-based entropy, which depends on the adjacency matrix, has been extensively investigated. Randic [16] applied eigenvalue multiplicity to distinguish different types of DNA structures and control protein synthesis in 2001. Ivanciuc [17–19] investigated the materials and spectra of molecular graphs. Sivakumar and Dehmer [14] proposed the entropy of eigenvalue-based modulus, and proved that eigenvalue-based modulus measures have a high recognition rate for molecular structures. Therefore, eigenvalue-based entropy [20–22], which relies on an adjacent matrix, is an important branch of multiple types of graph entropies. However, the research objects mentioned above are all undirected graphs [23], and the connection direction of nodes is not considered in real networks.

Real networks have more important direction properties than undirected networks. In 1968, Mowshowitz [24] investigated the entropy of digraphs, which is originally developed to obtain the entropy of digraphs [25, 26]. Since the matrix of the directed network is asymmetric

[27, 28], it is very difficult to investigate eigenvalue-based entropy in a directed network [29–31]. Moreover, investigations regarding eigenvalue-based entropy on a directed graph matrix are scarce.

### Research motivation

The research and application of eigenvalues based on the entropy of the directed graph is a necessary condition to fill this knowledge gap. Therefore, this article will conduct research from the following three aspects.

First, the definitions of the eigenvalue-based entropy of the adjacency, in-degree Laplacian, and in-degree signless Laplacian matrices in a directed network are provided herein. The eigenvalues of these matrices are typically complex numbers. Therefore, the corresponding eigenvalue-based entropy is classified as the real part entropy, imaginary part entropy, and modulus entropy.

Second, from typical models of complex networks, a model of directed network is proposed that considers the in-direction of node connections, and the eigenvalue-based entropy of the three matrices are calculated for the directed nearest-neighbour coupling, directed small-world, directed scale-free, and directed random networks.

Finally, by analysing simulation experiments on a real directed network, the results show that the eigenvalue-based entropy of the real directed network is between those of directed small-world and directed scale-free networks. Additionally, simulation results are provided to demonstrate the efficiency of the approach.

### Basic concept and terminology

Let $G = (V, E)$ be a finite undirected graph. The $V(G) = \{1, \cdots, n\}$ is the set of vertices and $E(G) = \{e_1, \cdots, e_m\}$ is the set of edges. Let $A(G)$ and $D(G)$ be the adjacency matrix and degree matrix of the graph $G$, respectively. The Laplace matrix is denoted as $L(G) = D(G) - A(G)$. The signless Laplacian matrix [32] is denoted as $Q(G) = D(G) + A(G)$.

In the literature [33], the adjacency matrix of the digraph is denoted as $A^-$, in-degree Laplacian matrix is denoted as $L^-$, in-degree signless Laplacian matrix is denoted as $Q^-$. Let $G = (V, E)$ be a digraph. The $V = \{1, 2, \cdots, n\}$ is the set of vertices. An ordered pair vertex $(v_j, v_i)$ is an edge of digraph, and the vertex $v_j$ walks to vertex $v_i$, which is denoted as $v_j \rightarrow v_i$. The in-degree and out-degree of a vertex $i$ are denoted as $d_i^-$ and $d_i^+$, respectively. The vertex $v_i$ in-degree sum is denoted as $\sum_{v_j \rightarrow v_i} d_i^-$. The in-degree matrix and the out-degree matrix are defined as $D^-$ and $D^+$, respectively. The adjacency matrix $A^-$ of digraph $G$ is denoted as:

$$A^- = \begin{cases} 0, & if \quad v_i = v_j, \\ 1, & if \quad v_j \rightarrow v_i, \\ 0, & otherwise. \end{cases}$$

For digraphs, the adjacency matrix is asymmetric. The Laplacian matrix $L^-$ of digraph $G$ is denoted as:

$$L^- = \begin{cases} d_i^-, & if \quad v_i = v_j, \\ -1, & if \quad v_j \rightarrow v_i, \\ 0, & otherwise. \end{cases}$$

The in-degree Laplacian matrix $Q^-$ of digraph $G$ is denoted as:

$$Q^- = \begin{cases} d_i^-, & if \quad v_i = v_j, \\ 1, & if \quad v_j \rightarrow v_i, \\ 0, & \text{otherwise.} \end{cases}$$

Let $\{\lambda_1, \lambda_2, \cdots, \lambda_n\}$, $\{\mu_1, \mu_2, \cdots, \mu_n\}$ and $\{q_1, q_2, \cdots, q_n\}$ be eigenvalue of the adjacency matrix, in-degree Laplacian matrix and in-degree signless Laplacian matrix in directed network, respectively. Since the asymmetry of the directed network matrix, most of its eigenvalue are complex numbers, and there are positive and negative numbers among them. We propose a novel entropy eigenvalue-based of the adjacency matrix, in-degree Laplacian matrix and in-degree signless Laplacian matrix. Let the real part and imaginary part entropy be *Re* and *Im*, respectively. The $|\lambda_j|$ is the absolute value of the *j* eigenvalue of the adjacency matrix. The $|\mu_j|$ is the absolute value of the *j* eigenvalue of the in-degree Laplacian matrix. The $|q_j|$ is the absolute value of the *j* eigenvalue of the in-degree signless Laplacian matrix. Herein, there is no special statement that a directed network is equivalent to a directed graph.

Next, we define the eigenvalue-based entropy of the three matrixes for directed networks.

1. The eigenvalue-based entropy of adjacency matrix for directed graph.

**Definition 0.1** *The entropy of real part is defined as*

$$I(Re(A^-)) = -\sum_{j=1}^{n} \frac{|Re(\lambda_j)|}{\sum\limits_{k=1}^{n}|Re(\lambda_k)|} |log \frac{|Re(\lambda_j)|}{\sum\limits_{k=1}^{n}|Re(\lambda_k)|}.$$

**Definition 0.2** *The entropy of imaginary part is defined as*

$$I(Im(A^-)) = -\sum_{j=1}^{n} \frac{|Im(\lambda_j)|}{\sum\limits_{k=1}^{n}|Im(\lambda_k)|} log \frac{|Im(\lambda_j)|}{\sum\limits_{k=1}^{n}|Im(\lambda_k)|}.$$

**Definition 0.3** *The entropy of modulus is defined as*

$$I(A^-) = -\sum_{j=1}^{n} \frac{|\lambda_j|}{\sum\limits_{k=1}^{n}|\lambda_k|} log \frac{|\lambda_j|}{\sum\limits_{k=1}^{n}|\lambda_k|}.$$

2. The eigenvalue-based entropy of in-degree Laplacian matrix.

**Definition 0.4** *The entropy of real part is defined as*

$$I(Re(L^-)) = -\sum_{j=1}^{n} \frac{|Re(\mu_j)|}{\sum\limits_{k=1}^{n}|Re(\mu_k)|} log \frac{|Re(\mu_j)|}{\sum\limits_{k=1}^{n}|Re(\mu_k)|}.$$

**Definition 0.5** *The entropy of imaginary part is defined as*

$$I(Im(L^-)) = -\sum_{j=1}^{n} \frac{|Im(\mu_j)|}{\sum\limits_{k=1}^{n}|Im(\mu_k)|} log \frac{|Im(\mu_j)|}{\sum\limits_{k=1}^{n}|Im(\mu_k)|}.$$

**Definition 0.6** *The entropy of modulus is defined as*

$$I(L^-) = -\sum_{j=1}^{n} \frac{|\mu_j|}{\sum_{k=1}^{n}|\mu_k|} log \frac{|\mu_j|}{\sum_{k=1}^{n}|\mu_k|}.$$

3. The eigenvalue-based entropy of in-degree signless Laplacian matrix.

**Definition 0.7** *The entropy of real part is defined as*

$$I(Re(Q^-)) = -\sum_{j=1}^{n} \frac{|Re(q_j)|}{\sum_{k=1}^{n}|Re(q_k)|} log \frac{|Re(q_j)|}{\sum_{k=1}^{n}|Re(q_k)|}.$$

**Definition 0.8** *The entropy of imaginary part is defined as*

$$I(Im(Q^-)) = -\sum_{j=1}^{n} \frac{|Im(q_j)|}{\sum_{k=1}^{n}|Im(q_k)|} log \frac{|Im(q_j)|}{\sum_{k=1}^{n}|Im(q_k)|}.$$

**Definition 0.9** *The entropy of modulus is defined as*

$$I(Q^-) = -\sum_{j=1}^{n} \frac{|q_j|}{\sum_{k=1}^{n}|q_k|} log \frac{|q_j|}{\sum_{k=1}^{n}|q_k|}.$$

An example is given and used to calculate the eigenvalue entropy of the real part, imaginary part and modulus of three matrices in a directed graph. Fig 1 shows a simple digraph $G(V, E)$ with 4 vertices and 5 arcs.

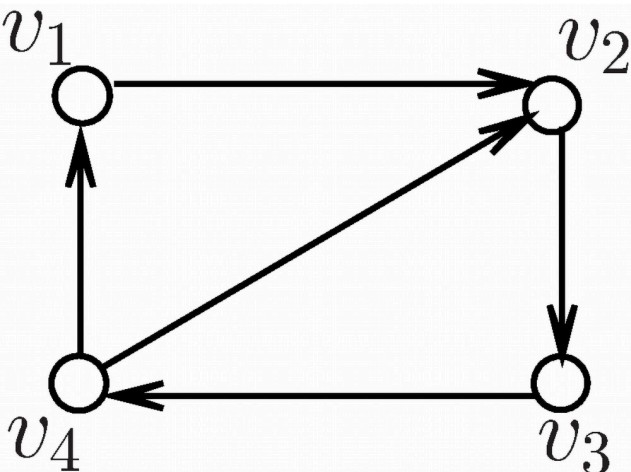

**Fig 1. Directed graph of four vertices.**

From Fig 1, we obtain the adjacency matrix, in-degree Laplacian matrix and in-degree sign-less Laplacian matrix as follows,

$$A^- = \begin{bmatrix} 0 & 1 & 0 & 0 \\ 0 & 0 & 1 & 0 \\ 0 & 0 & 0 & 1 \\ 1 & 1 & 0 & 0 \end{bmatrix}, \quad L^- = \begin{bmatrix} 1 & -1 & 0 & 0 \\ 0 & 2 & -1 & 0 \\ 0 & 0 & 1 & -1 \\ -1 & -1 & 0 & 1 \end{bmatrix}, \quad Q^- = \begin{bmatrix} 1 & 1 & 0 & 0 \\ 0 & 2 & 1 & 0 \\ 0 & 0 & 1 & 1 \\ 1 & 1 & 0 & 1 \end{bmatrix}$$

From Fig 1, we calculate eigenvalues of the three matrices, $\lambda_1 = 1.2207+0.0000i$, $\lambda_2 = -0.2481+1.0340i$, $\lambda_3 = -0.2481 - 1.0340i$, $\lambda_4 = -0.7245 +0.0000i$, where $\lambda_j(j = 1, 2, 3, 4)$ is the eigenvalue of adjacency matrix, and $i$ is an imaginary unit.

$\mu_1 = 0.0000+0.0000i$, $\mu_2 = 1.5000+0.8660i$, $\mu_3 = 1.5000 - 0.8660i$, $\mu_4 = 2.0000 +0.0000i$, where $\mu_j(j = 1, 2, 3, 4)$ is the eigenvalue of in-degree Laplacian matrix, and $i$ is an imaginary unit.

$q_1 = 0.3820+0.0000i$, $q_2 = 1.0000 - 0.0000i$, $q_3 = 1.000+0.0000i$, $q_4 = 2.6180+0.0000i$, where $q_j(j = 1, 2, 3, 4)$ is the eigenvalues of in-degree signless Laplacian matrix, and $i$ is an imaginary unit. The values above indicate that the eigenvalues are complex numbers. The eigenvalues appear as conjugate pairs, such as $\lambda_2$ and $\lambda_3$, $\mu_2$ and $\mu_3$, $q_2$ and $q_3$.

From Definition 0.1 to 0.9, we calculate the entropy of the eigenvalue in Fig 1 (please refer to Table 1).

Table 1 shows the nine eigenvalue-based entropy values of the three types of matrices for the directed network from Fig 1, let $I(Re(A^-))$, $I(Im(A^-))$ and $I(A^-)$ be the real part, imaginary part, and the modulus of the eigenvalue-based entropy on the adjacent matrix. Let $I(Re(L^-))$, $I(Im(L^-))$ and $I(L^-)$ be denoted by the real part, imaginary part, and the modulus of the eigen-value-based entropy on the in-degree Laplacian matrix. Let $I(Re(Q^-))$, $I(Im(Q^-))$ and $I(Q^-)$ be denoted by the eigenvalue-based entropies on the in-degree signless Laplacian matrix. As shown in Table 1, the numerical solution $I(Im)$ is eigenvalue-based entropy of the imaginary part in the three matrices. These results are consistent,

$$I(Im(A^-)) = I(Im(L^-)) = I(Im(Q^-)) = 0.6931.$$

This result indicates that node connections are the same direction in the digraph. Hence, the structural information can be captured by the eigenvalue-based entropy based on the three types of matrix in the directed network.

**Remark 0.1** *In the literature* [32], *if digraph is regular of a certain degree $d^-$, three matrices are the adjacency and in-degree Laplacian and in-degree signless Laplacian. The relationship between three matrices is as follow, so the adjacent spectrum is,*

$$[\lambda_1, \lambda_2, \cdots, \lambda_n], \tag{1}$$

Table 1. The eigenvalue-based entropy of three matricesfor the directed network of Fig 1 (n = 4).

| Matrix | Eigenvalue-based entropy | | |
|---|---|---|---|
| | The real part | The imaginary part | The modulus |
| adjacent matrix | $I(Re(A^-)) = 1.1718$ | $I(Im(A^-)) = $ **0.6931** | $I(A^-) = 1.3696$ |
| in-degree Laplacian matrix | $I(Re(L^-)) = 1.0889$ | $I(Im(L^-)) = $ **0.6931** | $I(L^-) = 1.0962$ |
| in-degree signless Laplacian matrix | $I(Re(Q^-)) = 1.1790$ | $I(Im(Q^-)) = $ **0.6931** | $I(Q^-) = 1.3723$ |

the Laplacian spectrum is,

$$[d^- - \lambda_1, d^- - \lambda_2, \cdots, d^- - \lambda_n],\tag{2}$$

*the signless Laplacian spectrum is,*

$$[d^- + \lambda_1, d^- + \lambda_2, \cdots, d^- + \lambda_n].\tag{3}$$

When the eigenvalues are complex numbers, the real part of eigenvalues is a real number, and in-degree is a real number. According to Eqs (1)–(3), the imaginary part entropy are equivalent for a regular digraph, i.e.

$$I(Im(A^-)) = I(Im(L^-)) = I(Im(Q^-)).$$

## Directed complex network model

We utilize the in-degree of the vertices to define the three matrices of the directed network. Let $\sum_{v_j \to v_i} d_i^-$ denote the total number of arcs. It is the sum of the in-degrees of vertex $v_i$ that walks from vertex $v_j$ to vertex $v_i$. In recent years, scholars have conducted empirical research through the analysis of computer technology networks, food networks, the world wide web, cell networks, circuit networks, etc. Directed network model has been proposed, and the characteristics and simple applications of these directed network models have been investigated. Schwartz [34] investigated the excesses of directed scale-free networks; Tadic [25] proposed a directed network model representing the www network; Ramezanpour [26] investigated a propagation process used in directed network [7] research. Murai [35] conducted a preliminary study on the spectrum properties [36] of a directed network. However, the modelling of directed network in the in-degree direction of $v_i$ is insufficient. The in-degree direction is from $v_j$ to $v_i$. Herein, we propose a novel directed complex network model that is constructed through algorithm improvement using a typically undirected complex network model.

## Directed random network model

To construct a directed random network [37], we regenerate models by using undirected idea of the Erdios and Renyi [15] in this paper, where the directions of the arcs are considered. Subsequently, a directed random network model is proposed. The construction process is as follows:

**Step 1.** Initially, set $n$ as the total number of nodes and random connection probability $p \in (0, 1)$.

**Step 2.** Randomly select different $t$ nodes from $n$ nodes as the arc-end.

**Step 3.** Randomly generate a number $p_1 \in (0, 1)$,

**Step 4.** If $p_1 > p$, select $r$ nodes in Step 3 as an arc-head connected by Step 2, and generate directed arcs.

**Step 5.** Repeat Steps 1–4 for each node $v_i$, and select different nodes only once.

The arc number of the directed random network is $p\binom{n}{r}$ and the directed network does not allow repetition arcs and loops.

## Directed small world network model

The undirected small-world network model is used to generate directed small-world network [38]. Construction process is as follows:

**Step 1.** In the initial directed nearest-neighbour network, set $n$ as the total number of nodes. Randomly generate the reconnection probability $p_1 \in (0, 1)$.

**Step 2.** Randomly select the $k$ nearest neighbour $v_{i+1+k}$ in the directed nearest-neighbour coupling network and random walk to any node $v_i$, and connect $v_{i+1+k} \rightarrow v_i$ to two nodes forming an arc.

**Step 3.** Repeat Step 2 until all $n$ different nodes are selected once.

**Step 4.** Generate a random number $p_1 \in (0, 1)$, if $p_1 \leq p$, then the arc will be randomised to reconnect, otherwise, the arc will not be reconnected.
Reconnection strategy: first, shift down the original arc-head and then randomly select another node as the arc-head from the unconnected nodes to connect with the original node.

**Step 5.** Until all nodes in the network are traversed.

## Directed scale-free network model

In 1999, Barabási and Albert [39] first proposed a network model derived from the dynamic evolution of growth and preferential connection mechanisms, empirically demonstrating the universal nature of a real network, where the number of nodes with large degrees is small in the network, whereas the number of nodes with small degrees is large in the network. In an undirected scale-free complex network [40, 41], the degree of a node obeys power-law distribution [42], $Z_{\text{deg}}(d) \propto d^{-\gamma}$, where $\gamma$ is the exponential value. The function $Z_{\text{deg}}(d)$ increases as the vertex degree $d$ decays slowly. We construct a directed scale-free complex network, where the in-degree of the node obeyed the power law distribution. The construction process of the directed scale-free network is as follows:

**Step 1.** Initially, set the number of network nodes before the network growth $m_0$; randomly specify the number of newly generated $m$ edges each time a new node is induced, and the network size after growth is recorded as $n$.

**Step 2.** Before the growth of the network (the number of nodes is $m_0$), randomly generate a number $p_1 \in (0, 1)$. When the probability is $p_1$, select a node $v_i$ as the arc-head and another node $v_j$ as the arc-end connection, which randomly connects $m_0$ nodes as a directed random network.

**Step 3.** Growth mechanism: based on Step 2, in executing $t = n-m_0$ time steps, add $s$ nodes in each time step, priority select $m$ nodes of existed to connect with the newly added node $s$, add $m$ arc in each time, and calculate the cumulative in-degree connection probability

$$q = \sum_{i=1}^{n}(d_i^-), q \in (0, 1)$$ of each node in the network. Let the total number of the network be

$n$. The network does not allow repetition arcs and loops.

**Step 4.** Preferential connection mechanism: In Step 3, $m$ nodes are selected from the existing nodes, and when the end of the arc is connected to the newly added node $s$, a new node $s$ is added in each time step based on the preferential probability $p_2 \in (0, 1)$, $p_2 = \dfrac{d_i^-}{\sum_{i=1}^{n} d_i^-} = \dfrac{d_i^-}{q}$.

The newly added nodes follow the mechanism of prioritising connections to known nodes to form a directed scale-free network.

### Directed nearest-neighbor coupling network model

The nearest-neighbour coupling network is a model that has been extensively investigated. In this paper, we construct a directed nearest-neighbour coupling network. The node of the network is arbitrarily selected to connect an arc with the nearest neighbour, and the direction of the arc is randomly selected. The procedure to construct process is as follows:

**Step 1.** Initially, set a number as the $k$th nearest neighbours, $n$ is the total number of nodes in the directed network.

**Step 2.** Randomly select the $k$ nearest-neighbour node in the network $v_{i+1+k}$ and walk to any node $v_i$, connect the node $v_{i+1+k}$ to $v_i$ forming an arc.

**Step 3.** Repeat Step 2 until all $n$ different nodes are selected once.

The following examples verify the effectiveness of the algorithm for constructed the model. The results show the directed networks of the directed nearest-neighbour coupling, directed small-world, directed scale-free, and directed random. They are shown in Figs 2 and 3.

### Simulation experiment and result analysis

To investigate the properties of the directed network model, a simulation experiment is performed based on the number of directed network nodes $n = 1000$ and the probability of node reconnection $p = 0.1 \sim 0.9$. For each node in the nearest-neighbor network, its neighbors are $k = 1, 3, 5, 7, 9$ and for each experimental result is the average result of 100 values. The simulation experiment conditions are divided into the following two categories:

1. Fixed number of nodes in a directed network.

(a)                                                                              (b)

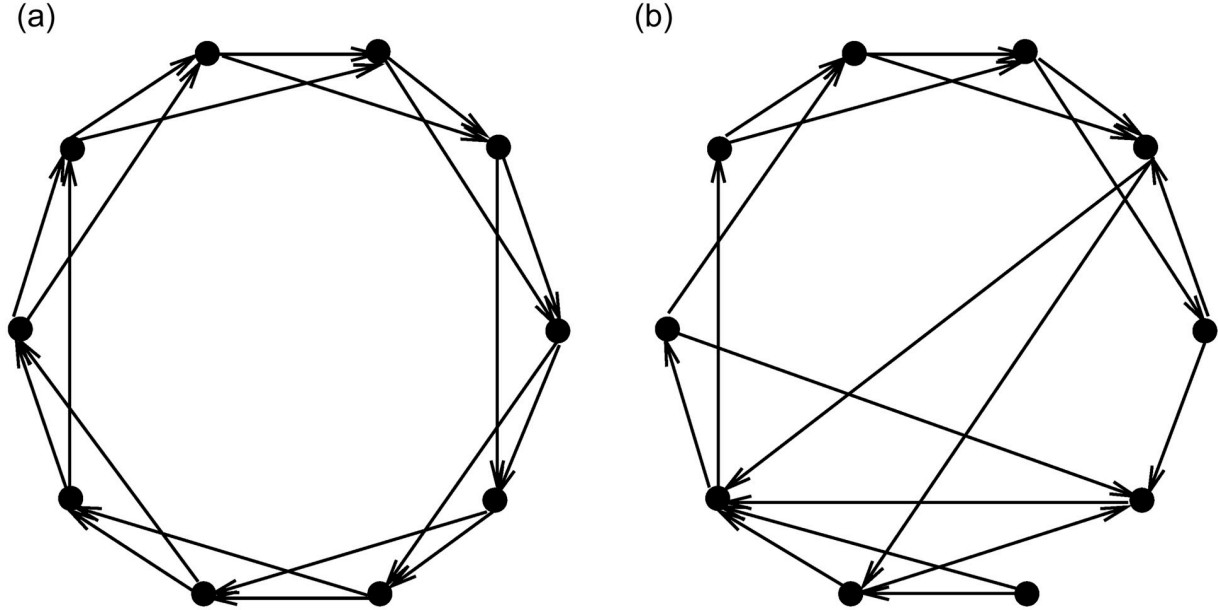

**Fig 2.** (a) directed nearest-neighbor coupled. (b) directed small-world network ($n = 10$, $k = 2$, $p = 0.1$).

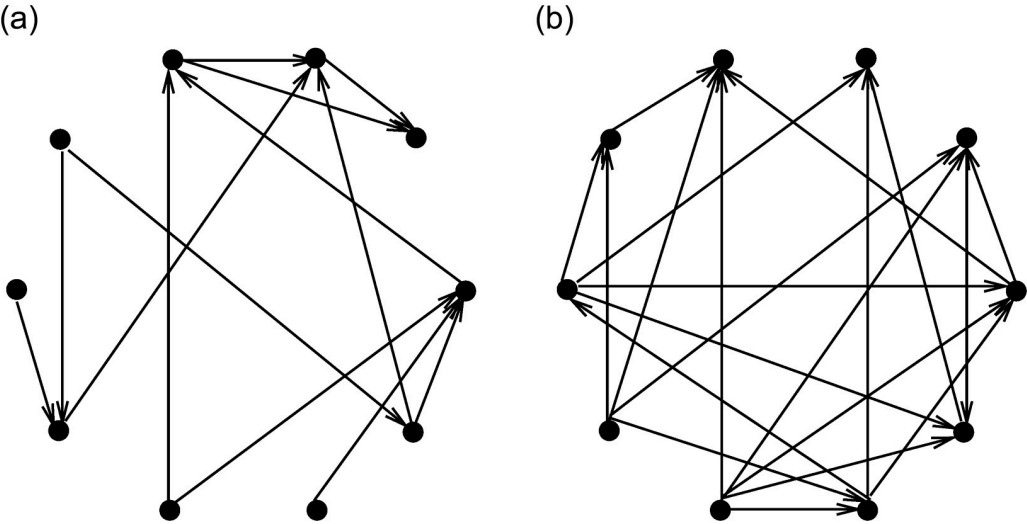

**Fig 3.** (a) directed scale-free network $n = 10$. (b) directed random network ($n = 10$, $p = 0.3$).

2. Fixed number of nodes in a directed network and the average in-degree $d_i^-$ of each node.

The results are shown in Tables 2–5. For the directed random, directed small-world, directed scale-free, and directed nearest-neighbour coupling networks, their eigenvalue-based entropy values are obtained from the first condition.

Based on information theory, entropy can reflect the irregular features of a network. The smaller the entropy value is, the more regular the network is, while the larger the entropy value is, the more irregular the network is. Based on the principle of entropy, we can assess the randomness and disorder of a network. The larger the eigenvalue-based entropy is in a directed network, the more dispersed the distribution of eigenvalue entropy is, thus the distribution of nodes is more equilibrium. When the entropy based on eigenvalue is smaller, the distribution of entropy is more concentrated. As a result, the distribution of nodes is more uneven.

The entropies of nine eigenvalues of adjacency matrix, in-degree Laplacian matrix and in-degree signless Laplacian matrix in directed random network are shown in Table 2. The values are obtained using the constructed directed random network. As shown in Table 2, the

**Table 2. The eigenvalue-based entropy of three matrices for directed random network.**

| Entropy | Reconnection probability $p$ | | | | |
|---|---|---|---|---|---|
| | **0.1** | **0.3** | **0.5** | **0.7** | **0.9** |
| $I(Re(A^-))$ | 6.6023 | 6.3972 | 6.5180 | 6.5698 | 6.6105 |
| $I(Im(A^-))$ | 6.5628 | 6.6307 | 6.6515 | 6.6535 | 6.6428 |
| $I(A^-)$ | 6.5652 | 6.6828 | 6.7310 | 6.7590 | 6.7757 |
| $I(Re(L^-))$ | 6.7492 | 6.7616 | 6.7455 | 6.7067 | 6.6281 |
| $I(Im(L^-))$ | 6.0982 | 6.1863 | 6.1739 | 6.0602 | 5.6040 |
| $I(L^-)$ | 6.7697 | 6.7618 | 6.7474 | 6.7074 | 6.6269 |
| $I(Re(Q^-))$ | 6.7494 | 6.7619 | 6.7458 | 6.7069 | 6.6282 |
| $I(Im(Q^-))$ | 6.1614 | 6.2073 | 6.1732 | 6.0242 | 5.5741 |
| $I(Q^-)$ | 6.7500 | 6.7621 | 6.7477 | 6.7077 | 6.6270 |

**Table 3. The eigenvalue-based entropy of three matrices for directed small-world network.**

| Entropy | Reconnection probability $p$ | | | | |
|---|---|---|---|---|---|
| | **0.1** | **0.3** | **0.5** | **0.7** | **0.9** |
| $I(Re(A^-))$ | 6.4492 | 6.5803 | 6.6571 | 6.6828 | 6.6784 |
| $I(Im(A^-))$ | 6.4160 | 6.5941 | 6.6588 | 6.6871 | 6.6754 |
| $I(A^-)$ | 6.5873 | 6.7375 | 6.8155 | 6.8230 | 6.8282 |
| $I(Re(L^-))$ | 6.8376 | 6.8316 | 6.8125 | 6.8050 | 6.7868 |
| $I(Im(L^-))$ | 6.2828 | 6.5347 | 6.6182 | 6.6181 | 6.6026 |
| $I(L^-)$ | 6.8609 | 6.8370 | 6.8250 | 6.8027 | 6.7989 |
| $I(Re(Q^-))$ | 6.8565 | 6.8372 | 6.8139 | 6.8056 | 6.7872 |
| $I(Im(Q^-))$ | 6.3182 | 6.5442 | 6.6231 | 6.6146 | 6.5969 |
| $I(Q^-)$ | 6.8555 | 6.8360 | 6.8250 | 6.8027 | 6.7996 |

**Table 4. The eigenvalue-based entropies of the three matrices for directed scale-free network.**

| Entropy | Network node $m_0 \rightarrow 1000$ | | | | |
|---|---|---|---|---|---|
| | **100 → 1000** | **200 → 1000** | **500 → 1000** | **700 → 1000** | **900 → 1000** |
| $I(Re(A^-))$ | 6.4721 | 6.3983 | 5.9254 | 6.1207 | 6.3287 |
| $I(Im(A^-))$ | 6.4656 | 6.3491 | 5.9090 | 6.1341 | 6.3411 |
| $I(A^-)$ | 6.6179 | 6.0769 | 6.0648 | 6.6269 | 6.7102 |
| $I(Re(L^-))$ | 5.6766 | 5.8970 | 6.2675 | 6.4271 | 6.6790 |
| $I(Im(L^-))$ | 6.3716 | 6.2206 | 5.7339 | 5.6420 | 5.7285 |
| $I(L^-)$ | 6.1932 | 6.1431 | 6.2678 | 6.6666 | 6.6279 |
| $I(Re(Q^-))$ | 5.6766 | 5.8971 | 6.2676 | 6.4271 | 6.6790 |
| $I(Im(Q^-))$ | 6.3716 | 6.1944 | 5.7251 | 5.5416 | 5.5762 |
| $I(Q^-)$ | 5.6842 | 6.1431 | 6.2678 | 6.6666 | 6.6279 |

probability of node reconnection is $p = 0.1 \sim 0.9$, therefore, the entropy value of the modular of in-degree Laplacian $I(L^-)$ is $6.7697 > 6.7618 > 6.7474 > 6.7074 > 6.6269$. The results show that the eigenvalue-based entropy values decrease gradually. Intuitively, the direction of the arc of the directed random network is random and diverse, and the entropy value of the imaginary part should increase. However, As shown in Table 2, the results show that the probability

**Table 5. The eigenvalue-based entropy of three matrices for directed nearest-neighbor coupled network.**

| Entropy | The nearest neighbor number $k$ | | | | |
|---|---|---|---|---|---|
| | **1** | **3** | **5** | **7** | **9** |
| $I(Re(A^-))$ | 6.7630 | 6.5354 | 6.3920 | 6.2916 | 6.1803 |
| $I(Im(A^-))$ | **6.7630** | **6.4923** | **6.3550** | **6.2588** | **6.1521** |
| $I(A^-)$ | 6.9078 | 6.6618 | 6.5210 | 6.4218 | 6.3115 |
| $I(Re(L^-))$ | 6.6009 | 6.7883 | 6.8334 | 6.8538 | 6.8695 |
| $I(Im(L^-))$ | **6.7630** | **6.4923** | **6.3550** | **6.2588** | **6.1521** |
| $I(L^-)$ | 6.7630 | 6.8497 | 6.8713 | 6.8872 | 6.8888 |
| $I(Re(Q^-))$ | 6.6009 | 6.8328 | 6.8644 | 6.8773 | 6.8867 |
| $I(Im(Q^-))$ | **6.7630** | **6.4923** | **6.3550** | **6.2588** | **6.1521** |
| $I(Q^-)$ | 6.7630 | 6.8347 | 6.8620 | 6.8745 | 6.8842 |

of node reconnection increases and the entropy of imaginary part of eigenvalue decreases. As shown in Table 2, with a fixed reconnection probability, when the reconnection probability is $p = 0.1$, the modular and real part entropies of the three types of matrices do not differ significantly. The value $I(Re(*)) \simeq I(*) > I(Im(*))$ is larger than the entropy value of the imaginary part in the corresponding matrix, where $*$ is the wildcard of $A^-$, $L^-$ and $Q^-$. This indicates increasingly irregular connections between nodes in the directed random network, and the increasing equilibrium in the distribution of directed arcs.

The entropies of nine eigenvalues of adjacency matrix, in-degree Laplacian matrix and in-degree signless Laplacian matrix in the directed small-world network are shown in Table 3. The probability of node reconnection is $p = 0.1 \sim 0.9$. When the probability increases with the increase of entropy, the results show that the direction of the arc tends to diverge and become erratic When the reconnection probability $p = 0.1$, the real part and modular entropy of these matrix exhibit $I(Re(L^-)) > I(Re(A^-))$ and $I(L^-) > I(A^-)$ and $I(Re(Q^-)) > I(Re(A^-))$ and $I(Q^-) > I(A^-)$, respectively. The imaginary part entropy exhibits $I(Im(L^-)) < I(Im(A^-))$ and $I(Im(Q^-)) < I(Im(A^-))$. The results show that the directed small-world network is more random, and the direction of arc tends to be centralized. Through the analysis of the above results, it is found that the directed small-world network is a process from the directed regular network to the directed random network.

The eigenvalue-based entropies of the three matrices for directed scale-free network are shown in Table 4. When the initial nodes are a percentage of the overall network nodes from 0% $\sim$ 50%, the entropy value decreased from 6.4721 to 5.9254. When the node ratio is 50% $\sim$ 90%, the entropy value increased from 5.9254 to 6.3287. Hence, the result shows that the node connection is in the priority select probability $p_2$ increases, and the directions of arcs become increasingly concentrated. Therefore, the entropy value decreases gradually. The entropy value fluctuates with the node degree obeying the power law, which is consistent with the structural characteristics of directed scale-free networks. When the node ratio is between 70% and 90%, the entropy value will increase. When the network nodes are in a certain proportion, the directed network appears to be chaotic and irregular. As shown in Table 4, $I(Re(A^-)) > I(Re(L^-)) > I(Re(Q^-))$, $I(Im(A^-)) > I(Im(L^-)) > I(Im(Q^-))$. The result $I(Im(A^-)) < I(Im(L^-)) < I(Im(Q^-))$, $I(A^-)) < I(L^-)) < I(Q^-))$ indicates that as the connected nodes of the directed network increased with the priority select probability $p_2$, the directed scale-free network nodes obey power law distribution.

Table 5 shows the directed nearest-neighbor coupling network. When the nearest neighbour $k = 9$, $I(Im(A^-)) = I(Im(L^-)) = I(Im(Q^-)) = 6.1521$, the values of the imaginary part entropy are identical. This verifies that the directions of the node connections are consistent in the directed nearest-neighbour coupling network. By analysing the entropy value of the imaginary part in Table 5, When the number of neighbors $k$ in the nearest-neighbour coupling network increases, the entropy decreases: $6.7630 > 6.4923 > 6.3550 > 6.2588 > 6.1521$. This indicates that the network direction become increasingly concentrated. The experimental simulation results of the directed nearest-neighbour network are consistent with the theoretical analysis.

To sum up, the values of nine eigenvalue-based entropies of three types of matrices are obtained experimentally in this study. Tables 2–5 show that eigenvalue-based entropy can effectively quantify the structural characteristics of the directed network model.

According to Tables 2–5, a better visualization structure is shown in Figs 4 and 5.

Figs 4 and 5 show the features of (a) the directed random network. (b) directed small-world network. (c) directed scale-free network. (d) directed nearest-neighbour coupling network.

The network nodes at the start of the scale-free directed network shown in Fig 5 are normalised: proportion of Horizontal coordinate $= \frac{m_0}{\text{total number of nodes in the network}}$.

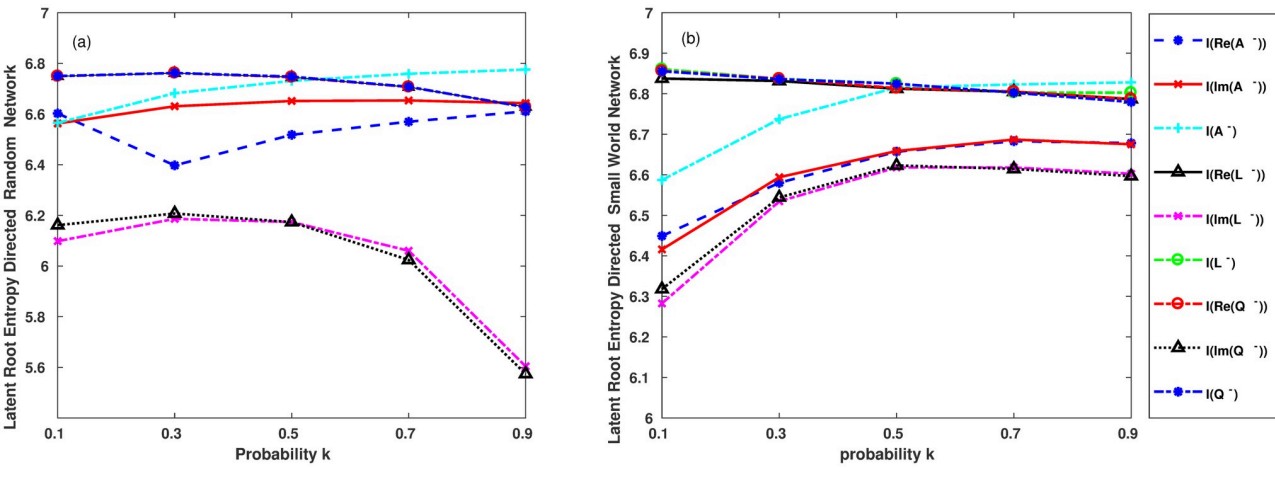

**Fig 4.**

As shown in Figs 4 and 5, the characteristics of the eigenvalue-based entropy of the directed network are as follows:

First, we can obtain the entropy values as the fluctuations in the directed scale-free network, as shown in Fig 5(c).

Second, in Figs 4 and 5, by comparing the entropy values of the imaginary parts $I(Im(A^-))$ and $I(Im(L^-))$ and $I(Im(Q^-))$, we find that the entropy of the imaginary part of the three matrices in the directed nearest-neighbour coupling network is consistent. This indicates that the directions of the arcs are identical in the directed nearest-neighbour coupling network. However, the eigenvalue-based entropies of the other directed networks do not exhibit this feature.

Third, we compare and observe the entropy of the imaginary part of the three matrices for the directed scale-free network shown in Fig 5(c). When the initial network nodes are changed from 10% to 50%, $I(Im(A^-))$ reached a low value, and $I(Im(A^-))$ increased gradually from 50% to 90%. However, in the 10% ∼ 90% range, $I(Im(L^-))$ and $I(Im(Q^-))$ decreases gradually. It

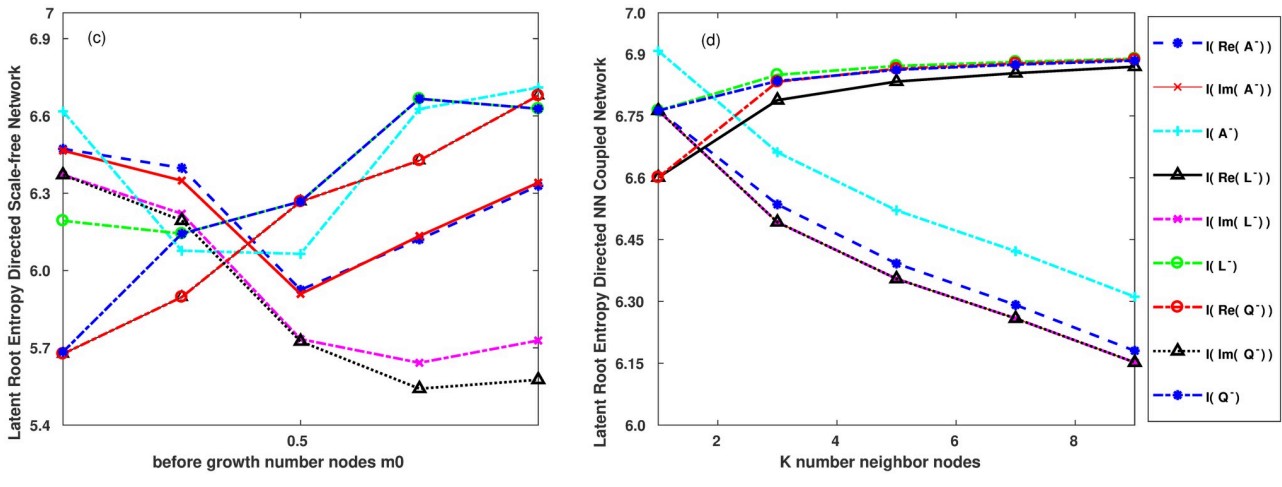

**Fig 5.**

**Table 6. The Eigenvalue-based entropy of three matrices for the directed generated network on the average in-degree.**

| Entropy | directed network | | | |
|---|---|---|---|---|
| | random | small world | scale-free | NN coupling |
| $I(Re(A^-))$ | 6.5654 | 6.5295 | 4.9618 | 6.6479 |
| $I(Im(A^-))$ | 6.5690 | 6.4838 | 5.0305 | **6.5948** |
| $I(A^-)$ | 6.6440 | 6.6254 | 5.1702 | 6.7629 |
| $I(Re(L^-))$ | 6.7429 | 6.7002 | 6.1542 | 6.7361 |
| $I(Im(L^-))$ | 6.5195 | 6.4588 | 4.8956 | **6.5948** |
| $I(L^-)$ | 6.7556 | 6.7334 | 6.1552 | 6.7447 |
| $I(Re(Q^-))$ | 6.7434 | 6.7203 | 6.1548 | 6.7892 |
| $I(Im(Q^-))$ | 6.5161 | 6.4922 | 4.8941 | **6.5948** |
| $I(Q^-)$ | 6.7562 | 6.7269 | 6.1559 | 6.8547 |

shows that the node direction changes concentrated when the in-degree of the node increases in the scale-free network. Moreover, it can be seen that the eigenvalue-based entropy of the in-degree Laplacian matrix and the in-degree unsigned Laplacian matrix can better reflect the characteristics of power-law in the directed scale-free network.

For the fixed number of nodes and arcs, that is, under the second experimental condition, the entropy results of the eigenvalues of the directed network with average penetration $< d_i^- = 100 >$ are shown in Table 6 and Fig 6.

Table 6 shows the average in-degree $< d_i^- = 100 >$ of the four generated models of the directed complex network, and the eigenvalue-based entropy of the three matrices for $n = 1000$. As shown in Table 6, the imaginary part entropy of the eigenvalue-based entropy in the directed nearest-neighbour coupling network, i.e. $Im(A^-)$, $Im(L^-)$ and $Im(Q^-)$ are consistent, whereas the eigenvalue-based entropy for the directed scale-free network change significantly.

Fig 6 shows the eigenvalue-based entropy results when the average in-degree is $< d_i^- = 100 >$. The columns of each cluster in Fig 6 show eigenvalue-based entropy of the (a) directed random network, (b) directed small world network, (c) directed scale-free network, (d) directed nearest neighbor coupling network, which corresponding to adjacency and the in-degree Laplacian and in-degree signless Laplacian matrices.

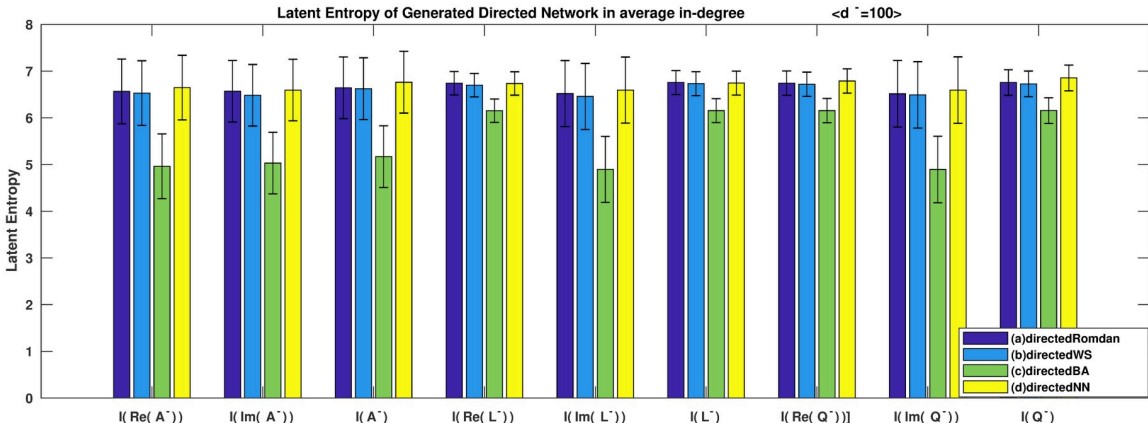

**Fig 6. Eigenvalue-based entropy of three matrix for the directed construction network: $< d_i^- = 100 >$, $n = 1000$.**

**Table 7. The Eigenvalue-based entropy of three matrices for the European mailnetwork.**

| Matrix | Eigenvalue-based entropy | | |
|---|---|---|---|
| | **The real part** | **The imaginary part** | **The modulus** |
| adjacent matrix | $I(Re(A^-)) = 6.1620$ | $I(Im(A^-)) = 6.3017$ | $I(A^-) = 6.2457$ |
| in-degree Laplacian matrix | $I(Re(L^-)) = 6.3814$ | $I(Im(L^-)) = 5.4244$ | $I(L^-) = 6.3814$ |
| in-degree signless Laplacian matrix | $I(Re(Q^-)) = 6.4194$ | $I(Im(Q^-)) = 5.2737$ | $I(Q^-) = 6.4194$ |

Fig 6 shows the nine eigenvalue-based entropies and the standard deviations of the four models of directed complex networks. The pillars in Fig 6 represent the eigenvalue-based entropies, and the whiskers in Fig 6 represent the standard deviations of the eigenvalue-based entropies. The standard deviation represents the degree of dispersion between the eigenvalue-based entropy of the directed network and its mean value.

## Eigenvalue-based entropy of real complex network

To prove the efficiency of the model, we investigate the real directed network. The dataset of the real directed network are from the data [43] of a large European research institution. The dataset contained 1,005 member nodes and 25,571 arcs. Table 7 shows the eigenvalue-based entropy of the real directed network. It is convenient to compare them with a real directed network dataset, which the number of nodes in the constructed network model is selected $n = 1000$.

The columns of each cluster in Fig 7 show eigenvalue entropy generated of the (*a*) directed random network, (*b*) directed small world network, (*c*) directed scale-free network, (*d*) directed nearest neighbor coupling network, (*e*) real directed network, which corresponding to adjacency and the in-degree Laplacian and in-degree signless Laplacian matrices. Furthermore, Fig 7 shows the eigenvalue-based entropy of the three matrices for directed construction networks *vs*. the real directed network. Fig 7. The eigenvalue-based entropy of three matrices for the directed construction networks *vs*. real directed network.

In addition, Fig 7 shows the eigenvalue-based entropies of the constructed directed and real directed networks under the condition of average in-degrees $< d_i^- = 100 >$ and $n = 1000$. The entropy of real directed European E-mail network is in the middle of directed small world

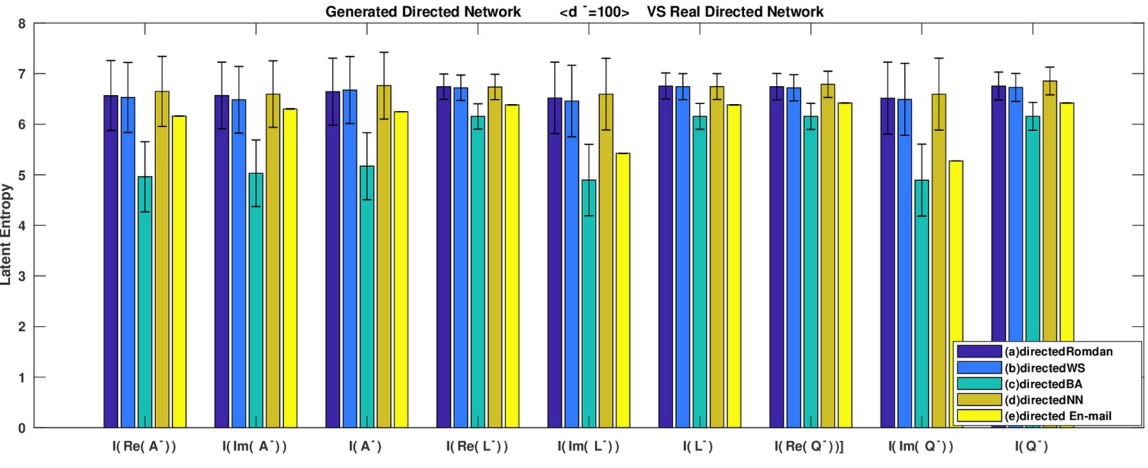

**Fig 7.**

network and directed scale-free network, so the real directed network has the structural characteristics of directed small world network and directed scale-free network.

## Conclusion

In this paper, we investigate the directional characteristic of node connections in directed complex networks by modelling directed nearest-neighbour coupling, directed small-world, directed scale-free, and directed random networks. We define the entropy of the eigenvalues of the adjacency matrix, in-degree Laplacian matrix, and in-degree signless Laplacian matrices in the directed network. Through the entropy of the eigenvalues of the three matrices, the directional characteristics of the directed network can be captured. i.e. The simulation results show that the entropy of the eigenvalues of the directed complex network can described the structural characteristics of the directed network, and the real directed complex network has characteristics of small world and scale-free. The Definitions and methods demonstrate the effectiveness of eigenvalue-based entropy of the adjacency, in-degree Laplacian, and in-degree signless Laplacian matrices. It can capture the structural characteristics of directed network, and the research results can be applied to other real directed networks.

## Supporting information

**S1 Text.**
(TXT)

**S2 Text.**
(TXT)

## Acknowledgments

We thank the editors and the anonymous reviewers for their professional and valuable suggestions. We thank the Tibetan Information Processing and Machine Translation Key Laboratory of Qinghai Province (Grant No. 2020-ZJ-Y05) and the Key Laboratory of Tibetan Information Processing Ministry of Education and Tibetan Information Processing Engineering Technology and Research Center of Qinghai Province.

## Author Contributions

**Data curation:** Jing Liang.

**Formal analysis:** Haixing Zhao.

**Methodology:** Yan Sun.

**Visualization:** Xiujuan Ma.

**Writing – original draft:** Yan Sun.

**Writing – review & editing:** Haixing Zhao.

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
