## [Decision Letter · Decision Letter 0]

4 Mar 2021

PONE-D-21-02058

Eigenvalue-based Entropy in Directed Complex Networks

PLOS ONE

Dear Dr. sun,

Thank you for submitting your manuscript to PLOS ONE. After careful consideration, we feel that it has merit but does not fully meet PLOS ONE’s publication criteria as it currently stands. Therefore, we invite you to submit a revised version of the manuscript that addresses the points raised during the review process.

According to the reviewers' comments, suggestion and concerns as well as my own reading of this manuscript, I think the manuscript has some major problems on writing, which directly influences the understanding. 

Meanwhile, some claims need to prove or explain with more details. 

We look forward to receiving your revised manuscript.

Kind regards,

Weinan Zhang

Academic Editor

PLOS ONE

Journal Requirements:

"Specify the role(s) played."

Reviewers' comments:

Reviewer's Responses to Questions

**Comments to the Author**

1. Is the manuscript technically sound, and do the data support the conclusions?

Reviewer #1: Yes

Reviewer #2: Yes

2. Has the statistical analysis been performed appropriately and rigorously? 

Reviewer #1: I Don't Know

Reviewer #2: Yes

3. Have the authors made all data underlying the findings in their manuscript fully available?

Reviewer #1: Yes

Reviewer #2: Yes

4. Is the manuscript presented in an intelligible fashion and written in standard English?

Reviewer #1: No

Reviewer #2: Yes

5. Review Comments to the Author

Reviewer #1: REPORT ON “Eigenvalue-based Entropy in Directed Complex Networks”

YAN SUN, HAIXING ZHAO, JING LIANG, AND XIUJUAN MA

This paper based on typical models of complex networks, a model of a directed network is proposed that considers the in-direction of connections, and the entropy of eigenvalues of three matrices for a directed network are defined and investigated: the adjacency matrix, in-degree Laplacian matrix, and in-degree signless Laplacian. The entropy of the three matrix eigenvalues of the directed nearest-neighbour coupling, directed small-world, directed scale-free, and directed random networks is calculated. By conducting simulation experiments on a real directed network, the results show that the eigenvalue-based entropy of the realistic directed network is between those of the directed small-world and directed scale-free networks.

This is an interesting manuscript. However, from the linguistic viewpoint the paper is poorly written and hard to read. Therefore it needs a deep revision. I invite the authors to full revise the english and the exposition of the paper (not only following my comments below). Please find below some minor mathematical suggestions, notation, style, and language issues that should be corrected, along with a list of typos.

Page 2, Line 27 ``coming from ".

Page 2, Line 34 write ``$\\ldots$, which based on $\\ldots$"

Page 3, Line 64 ``$\\ldots$ and$\\ldots$are$\\ldots$"

Page 4, Line 143 write ``Definition 0.1~0.9 "

Page 5, Line 124-125 There are no predicate verbs in this sentence, so it is not a complete sentence.

This paper shows that “the imaginary part entropy $I(Im(A^{-}))=I(Im(L^{-}))=I(Im(Q^{-}))$”. In theory, can this precise conclusion be proved? The referee suggests them consider the theoretical proof .

Reviewer #2: The paper investigates the directional characteristic of a node connection in a directed complex network, and exploits eigenvalue-based entropy to capture the directional characteristics of the directed network. The paper is well written, and with great values for publication.

6. PLOS authors have the option to publish the peer review history of their article (what does this mean?). If published, this will include your full peer review and any attached files.

Reviewer #1: No

Reviewer #2: No

---

## [Author Response · Author response to Decision Letter 0]

11 Mar 2021

1 We have carefully showing my changes by highlighting them and uploaded as a supporting information file.

2 We have a clean of the edited manuscript according to review letter. 

3 We have rechecked the paper and corrected any mistakes.

4 We upload my manuscript file in PDF format and attach your .tex file as “other.”

5 We upload a minimal data set as a Supporting Information file, to a public repository as Dryad. 

6 We have amended the list of affiliation's to include details for affiliations 3 and 4.

7 We have removed my figures/ from within the manuscript file, leaving only the individual TIFF/EPS image files. 

8 Project supported by the National Natural Science Foundation of China (Grant Nos.11661069,61663041), the Science and Technology Plan of Qinghai Province, China(Grant No.2019-ZJ-7012)

9 We ensure that my refer to Table 7 in my text, 

 Table 7 is calculated by Stanford http://snap.stanford.edu/data/email.

---

## [Decision Letter · Decision Letter 1]

7 May 2021

Eigenvalue-based Entropy in Directed Complex Networks

PONE-D-21-02058R1

Dear Dr. sun,

We’re pleased to inform you that your manuscript has been judged scientifically suitable for publication and will be formally accepted for publication once it meets all outstanding technical requirements.

Kind regards,

Weinan Zhang

Academic Editor

PLOS ONE

Additional Editor Comments (optional):

Although the revision addresses the main concerns of the reviewers, there are still some parts that need to be explained in more details.

Meanwhile, a proofread is also needed.

Reviewers' comments:

Reviewer's Responses to Questions

**Comments to the Author**

1. If the authors have adequately addressed your comments raised in a previous round of review and you feel that this manuscript is now acceptable for publication, you may indicate that here to bypass the “Comments to the Author” section, enter your conflict of interest statement in the “Confidential to Editor” section, and submit your "Accept" recommendation.

Reviewer #1: (No Response)

Reviewer #2: All comments have been addressed

2. Is the manuscript technically sound, and do the data support the conclusions?

Reviewer #1: Yes

Reviewer #2: Yes

3. Has the statistical analysis been performed appropriately and rigorously? 

Reviewer #1: Yes

Reviewer #2: Yes

4. Have the authors made all data underlying the findings in their manuscript fully available?

Reviewer #1: Yes

Reviewer #2: Yes

5. Is the manuscript presented in an intelligible fashion and written in standard English?

Reviewer #1: (No Response)

Reviewer #2: Yes

6. Review Comments to the Author

Reviewer #1: (No Response)

Reviewer #2: This paper proposed a new kind of definition of eigenvalue-based entropy for directed graph. It expends the entropy to complex field. With this new metric, people can model the random characteristics of directed complex networks with a new respect.

The paper has been greatly improved based on the former version.

7. PLOS authors have the option to publish the peer review history of their article (what does this mean?). If published, this will include your full peer review and any attached files.

Reviewer #1: No

Reviewer #2: No

---

## [Editor Report · Acceptance letter]

25 May 2021

PONE-D-21-02058R1 

Eigenvalue-based Entropy in Directed Complex Networks 

Dear Dr. Sun:

I'm pleased to inform you that your manuscript has been deemed suitable for publication in PLOS ONE. Congratulations! Your manuscript is now with our production department. 

Kind regards, 

on behalf of

Dr. Weinan Zhang 

Academic Editor

PLOS ONE